# Harnessing Dataset Cartography for Improved Compositional Generalization in Transformers

**Osman Batur İnce [1,2]**    **Tanin Zeraati [3]**    **Semih Yagcioglu [4]**
**Yadollah Yaghoobzadeh[3,5]**    **Erkut Erdem[4]**    **Aykut Erdem[1,2]**

[1] Koç University, KUIS AI Center  [2] Koç University, Computer Engineering Department
[3] University of Tehran, Iran  [4] Hacettepe University, Computer Engineering Department
[5] Tehran Institute for Advanced Studies, Khatam University, Iran

oince22@ku.edu.tr    tzeraati@ut.ac.ir    semih.yagcioglu@hacettepe.edu.tr
y.yaghoobzadeh@ut.ac.ir    erkut@cs.hacettepe.edu.tr    aerdem@ku.edu.tr

## Abstract

Neural networks have revolutionized language modeling and excelled in various downstream tasks. However, the extent to which these models achieve *compositional generalization* comparable to human cognitive abilities remains a topic of debate. While existing approaches in the field have mainly focused on novel architectures and alternative learning paradigms, we introduce a pioneering method harnessing the power of *dataset cartography* (Swayamdipta et al., 2020). By strategically identifying a subset of compositional generalization data using this approach, we achieve a remarkable improvement in model accuracy, yielding enhancements of up to 10% on CFQ and COGS datasets. Notably, our technique incorporates dataset cartography as a curriculum learning criterion, eliminating the need for hyperparameter tuning while consistently achieving superior performance. Our findings highlight the untapped potential of dataset cartography in unleashing the full capabilities of compositional generalization within Transformer models. Our code is available at https://github.com/cyberiada/cartography-for-compositionality.

## 1 Introduction

In recent years, deep learning methods and machine learning infrastructure have made remarkable progress, enabling models to surpass human-level performance in numerous tasks. Natural language processing (NLP) is at the forefront of this progress. Models based on Transformers (Vaswani et al., 2017) such as BERT (Devlin et al., 2019) and benchmarks like SuperGLUE (Wang et al., 2019) led to significant advancements in language modeling and various downstream tasks. However, there is an ongoing debate on whether these models exhibit compositional generalization (Fodor and Pylyshyn, 1988; Smolensky, 1988; Marcus, 2001; Lake and Baroni, 2017).

Compositional generalization refers to the ability of a model to combine known parts of a sentence, such as primitive tokens, to generate novel compositions of these primitive elements. It is considered a fundamental aspect of human cognition and linguistics (Fodor and Pylyshyn, 1988). In addition to its human aspect, compositional generalization is also crucial for enhancing the robustness and practical use of deep learning models. Efforts to understand and improve the compositional generalization abilities of models have gained significant attention lately. Researchers have recently explored techniques such as compositional data augmentation (Andreas, 2020; Qiu et al., 2022), meta-learning (Lake, 2019), and structural priors (Russin et al., 2020). Additionally, the importance of architectural modifications to capture compositional structures more effectively, such as attention mechanisms (Li et al., 2019) and hierarchical structures (Weißenhorn et al., 2022) have been investigated recently. In another direction, there is also an increasing interest in studying the compositional generalization abilities of Transformers (Ontanon et al., 2022; Csordás et al., 2021; Dziri et al., 2023).

In this study, we take a distinct approach and utilize *dataset cartography* (Swayamdipta et al., 2020) to explore how training dynamics can improve the compositional generalization abilities of Transformers. Dataset cartography is a recently proposed technique that quantifies the variability and confidence associated with instances during training, capturing their ambiguity and difficulty, thereby representing the *informational value* of each training sample. Swayamdipta et al. (2020) demonstrated that it could be used to improve out-of-distribution (OOD) generalization in models for classification-based natural language inference (NLI) tasks. As compositional generalization is inherently an OOD task, we hypothesize that harnessing dataset cartography in compositional generalization can provide new insights.

Diverging from the original cartography setup, we focus on language generation tasks for the systematic generalization problem and propose an experimental setting to apply dataset cartography to a generative task. Initially, we train a sequence-to-sequence (seq2seq) Transformer model using the complete training set for only a few epochs. Throughout the training, the dynamics of each instance are observed and recorded separately. Next, we utilize these stored training dynamics to build a curriculum and create a reduced training set by selecting specific samples to fully train the model.

Our experimental setup has notable challenges beyond the compositional generalization setting, distinguishing it from the setup originally used in Swayamdipta et al. (2020). Instead of relying on crowdsourced datasets that are prone to errors and heavily reliant on data quality, we utilize synthetically generated datasets, namely CFQ (Keysers et al., 2020) and COGS (Kim and Linzen, 2020), which are free from such limitations. Moreover, these datasets are relatively smaller in size, making it challenging to achieve performances *on par* with the 100% train set when using smaller subsets. Lastly, unlike Swayamdipta et al. (2020), we tackle the complexity of learning the task directly without using pre-trained models. This becomes even more pronounced as the datasets contain non-natural language, rendering pre-training less applicable and learning much harder. Finally, and more importantly, as we are dealing with language generation tasks, quantifying how hard a sequence is, is not straightforward. To address this, we base our evaluation by utilizing inverse perplexity (Inv PPL), CHIA (Bhatnagar et al., 2022), and BLEU (Papineni et al., 2002) as confidence measures, avoiding the overly strict exact matching strategy.

In summary, our paper makes the following key contributions: First, we introduce the novel use of dataset cartography as both a curriculum learning criterion and a sample selection strategy for enhancing compositional generalization. By leveraging dataset cartography, we enable models to deal effectively with the complexities of compositional tasks. Second, we thoroughly investigate the effectiveness of various confidence measures for sequences in extracting dataset cartography within the compositional generalization setting. This analysis provides insights into quantifying the difficulty of sequences and leads to the development of robust training strategies. Third, through extensive

analyses, we demonstrate the significant impact of leveraging training dynamics through dataset cartography on the compositional generalization capabilities of Transformer models. Our approach yields significant improvements of up to 10% on challenging CFQ and COGS datasets, highlighting the effectiveness of our proposed method.

## 2 Approach

### 2.1 Dataset Cartography

Swayamdipta et al. (2020) propose a visualization tool named **data maps** with two dimensions, **confidence** and **variability**, which characterizes the informativeness of training instances of a dataset with respect to a model. Confidence is calculated as the mean probability of the true label across epochs, whereas variability corresponds to the spread of confidence across epochs, using the standard deviation. Therefore, confidence ($\hat{\mu}_i$) and variability ($\hat{\sigma}_i$) is denoted as follows:

$$\hat{\mu}_i = \frac{1}{E} \sum_{e=1}^{E} p_{\theta^{(e)}}(y_i^* | \mathbf{x}_i) \qquad (1)$$

$$\hat{\sigma}_i = \sqrt{\frac{\sum_{e=1}^{E} (p_{\theta^{(e)}}(y_i^* | \mathbf{x}_i) - \hat{\mu}_i)^2}{E}} \qquad (2)$$

where $i$ denotes the instance, $E$ represents the total number of epochs, $\mathbf{x}_i$ is the input sequence, $y_i^*$ is the true label, $\theta^{(e)}$ corresponds to the set of model parameters at epoch $e$.

Data maps reveal three distinct regions: **ambiguous** instances (high variability), **easy-to-learn** instances (high confidence, low variability), and **hard-to-learn** instances (low confidence, low variability). Swayamdipta et al. (2020) experimented on multiple NLI datasets such as SNLI (Bowman et al., 2015) with pretrained models and reported three main findings: (i) Ambiguous regions contribute the most towards OOD generalization, (ii) Easy-to-learn regions play an important role in model optimization, (iii) Hard-to-learn regions often correspond to labeling errors.

### 2.2 Data Maps for Generative Tasks

The notions of confidence and variability in Eq. (1) and (2) are defined considering classification-based tasks, and thus not directly applicable to seq2seq models. Extending dataset cartography to machine translation, a generative task, Bhatnagar et al. (2022) propose the CHIA measure by following the intuition that an output sequence consists of

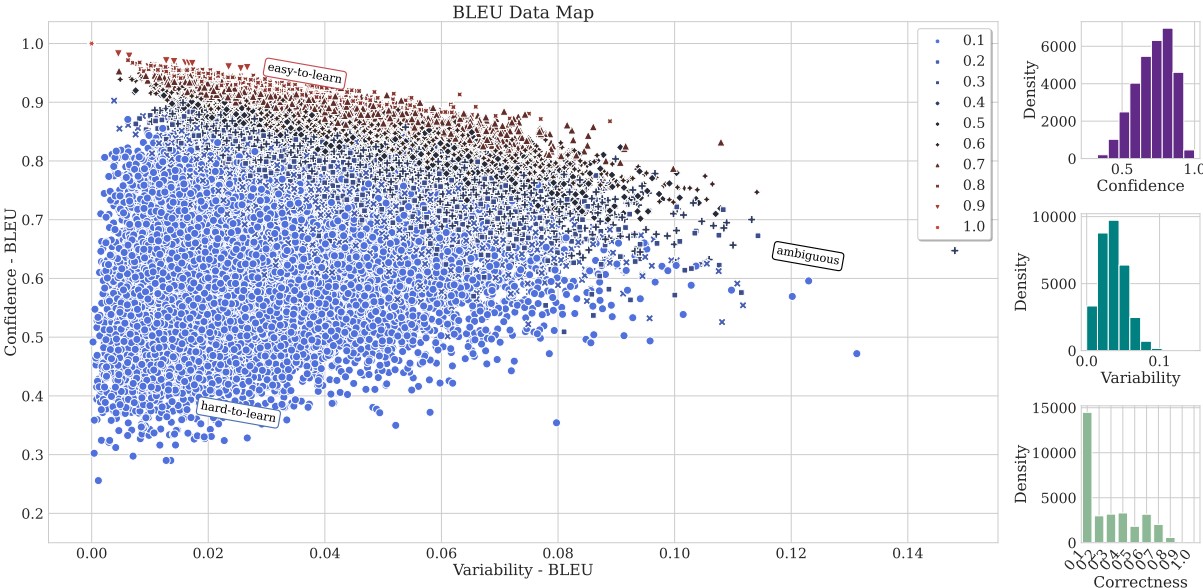

Figure 1: **Data map of CFQ train set** for the Transformer model based on BLEU measure (converge epoch 20). The *x*-axis shows the **variability** and the *y*-axis the **confidence**. The colors and shapes indicate the **correctness**.

a series of predictions. Instead of using the exact match, they take the arithmetic mean of gold token predictions over the sequence, defined as:

$$\hat{\mu}_i = \frac{1}{ET} \sum_{e=1}^{E} \sum_{t=1}^{T} p_{\theta^e} \left( y_{i_t}^* \mid \mathbf{x}_i \right) \qquad (3)$$

where $y_{i_t}^*$ corresponds to the $i$-th token of the groundtruth output sequence $\mathbf{y}_i$ of length $T$.

Here, it is important to note that Bhatnagar et al. (2022) do not use data maps to select a subset but instead use N-way translation corpora to choose instances that are most informative on all ways to select instances to annotate for low-resource languages. They showed that choosing instances based on a single-way translation decreases performance significantly, suggesting CHIA measure might not be the best choice for our experimental setting.

Similar to the CHIA score, we also consider inverse perplexity for the reason that high perplexity is an undesirable property. It is defined as the geometric mean of gold token predictions over the sequence, as given below:

$$\hat{\mu}_i = \frac{1}{E} \sum_{e=1}^{E} \prod_{t=1}^{T} \sqrt[T]{p_{\theta^e} \left( y_{it}^* \mid \mathbf{x}_i \right)} \qquad (4)$$

The geometric mean is much closer to the lowest probability in the sequence compared to the arithmetic mean used in CHIA, making inverse perplexity a more discriminative measure.

Additionally, we define a third measure based on BLEU (Papineni et al., 2002). In particular, BLEU measures the n-gram overlap between generated output and the ground truth, and we use the arithmetic mean of the BLEU score across epochs as a means to measure the confidence as follows:

$$\hat{\mu}_i = \frac{1}{E} \sum_{e=1}^{E} \text{BLEU}(\hat{\mathbf{y}}_{\mathbf{i}}^{(e)}, \mathbf{y}_{\mathbf{i}}) \qquad (5)$$

where $\hat{\mathbf{y}}_{\mathbf{i}}^{(e)}$ refers to the predicted sequence generated by the model parameters at epoch $e$, and $\mathbf{y}_{\mathbf{i}}$ denotes the ground truth sequence, respectively. A particular shortcoming of using BLEU is its computational and temporal expense due to decoding (for variability equations, see Appendix A.2).

The main motivation behind utilizing dataset cartography lies in selecting a subset of the training set and training the model on these subsets instead of the entire dataset. The selection process involves two key considerations: (1) Choosing the measure used to rank the examples, and (2) Determining the aspect of the measure scores for ranking (e.g., ambiguity). There are more hyperparameters such as subset ratio and subset combinations, until which epoch to take training dynamics into account (referred to as **convergence epoch**), from starting with which epoch training dynamics are considered (referred to as **min epoch**). Specifically, to identify the convergence epochs, we qualitatively examine loss convergence and generated data maps. Unlike

Swayamdipta et al. (2020), where authors utilize pre-trained models and consider training dynamics from the start of fine-tuning, our experimental setup involves randomly initialized models. Hence, considering training dynamics in the initial epochs while the model is not stable can result in noisy training dynamics (Swayamdipta et al., 2020), and can introduce selection biases based on data ordering. To simplify the process, we set the minimum epoch to 3 across all our training setups.

## 3 Experiments

### 3.1 Baselines

We benchmark with several baselines to demonstrate the enhancement in generalization performance through the selection of smaller, specifically chosen subsets. The most rudimentary of these baselines involves the selection of a random subset identical in size to the specifically chosen subset, along with the utilization of the entire original dataset for comparison, i.e. 100% of the original dataset. In the context of curriculum learning settings, we deem it necessary to establish another benchmark wherein no particular curriculum is employed. This serves as a baseline, facilitating the process of benchmarking for comparative purposes.

### 3.2 Datasets

We conduct our experiments on three compositional generalization datasets, CFQ (Keysers et al., 2020), COGS (Kim and Linzen, 2020), SMCalFlow-CS Simple (Meron, 2022, SMCS) datasets. CFQ and SMCS dataset has multiple splits. For CFQ, we utilize the MCD1 split. For SMCS, we utilize 16 and 32 splits, where the split numbers refer to the compositional example leak count into the training set. One challenge commonly encountered with compositional generalization datasets in the literature is the absence of validation splits. To ensure a fair comparison, we train all of our models with specified step counts, following the approach of Csordás et al. (2021).

To provide a better understanding of the datasets, let us consider specific examples from each. CFQ, being a synthetic text-to-SQL dataset, involves input samples such as "*Did a male film director produce and edit M1?*" with the corresponding target query being `SELECT count(*) WHERE {?x0 ns:film.producer.film M1 . ?x0 ns:film.editor.film M1 . ?x0 ns:people.person.gender m_05zppz}`. In the

| Dataset | #train | #test | Voc. size | Train len. | Test len. |
|---|---|---|---|---|---|
| CFQ | 95743 | 11968 | 181 | 29 / 95 | 30 / 103 |
| COGS | 24155 | 21000 | 871 | 22 / 153 | 61 / 480 |
| SMCS 16 | 25410 | 663 | 10738 | 107 / 103 | 30 / 59 |
| SMCS 32 | 25426 | 662 | 10738 | 107 / 103 | 30 / 59 |

Table 1: Dataset statistics showing sample counts, vocabulary size as the combined input and output vocabularies, and train and test length denoting the max. input/output length in the train and test set, respectively.

case of COGS, which is a synthetic semantic parsing task, an input sample could be "*A frog hopped*" and the corresponding target logical form is `frog(x1) AND hop.agent(x2, x1)`. For the natural semantic parsing dataset SMCS, an input sample is "*iam meet with smith , john and rodney*", and its output is `CreateEvent( AND( with_attendee( rodney ) , with_attendee( smith ) , with_attendee( john ) ) )`.

### 3.3 Experimental Setup

In our experiments, we employ the vanilla Transformer model (Vaswani et al., 2017). Recent studies have highlighted that the generalization capabilities of pre-trained Transformer models can be overestimated due to uncontrolled lexical exposure (Kim et al., 2022; An et al., 2023). We adopted the publicly available PyTorch codebase provided by Csordás et al. (2021) to implement our model. Each experiment is executed on a single Tesla T4 GPU. We employ a whitespace tokenizer for all datasets, considering that the targets in these datasets are not expressed in natural language. We also experiment with Bi-LSTM with attention (Bahdanau et al., 2015) on the COGS dataset.

In a similar way to Swayamdipta et al. (2020), we show the data maps generated for CFQ based on BLEU and COGS based on Inv PPL in Figure 1 and Figure 2, respectively. For better visualizations, we only plot randomly sampled 33% of the training set. Considering a larger number of training epochs compared to Swayamdipta et al. (2020), we divide the correctness scores into 10 bins for better granularity and informative visualizations. As we discussed earlier, we use three distinct confidence measures, inverse perplexity (Inv PPL), CHIA (Bhatnagar et al., 2022), and BLEU (Papineni et al., 2002). The missing data maps are given in Appendix A.7.

We explore two different setups to assess the effectiveness of leveraging data maps in improv-

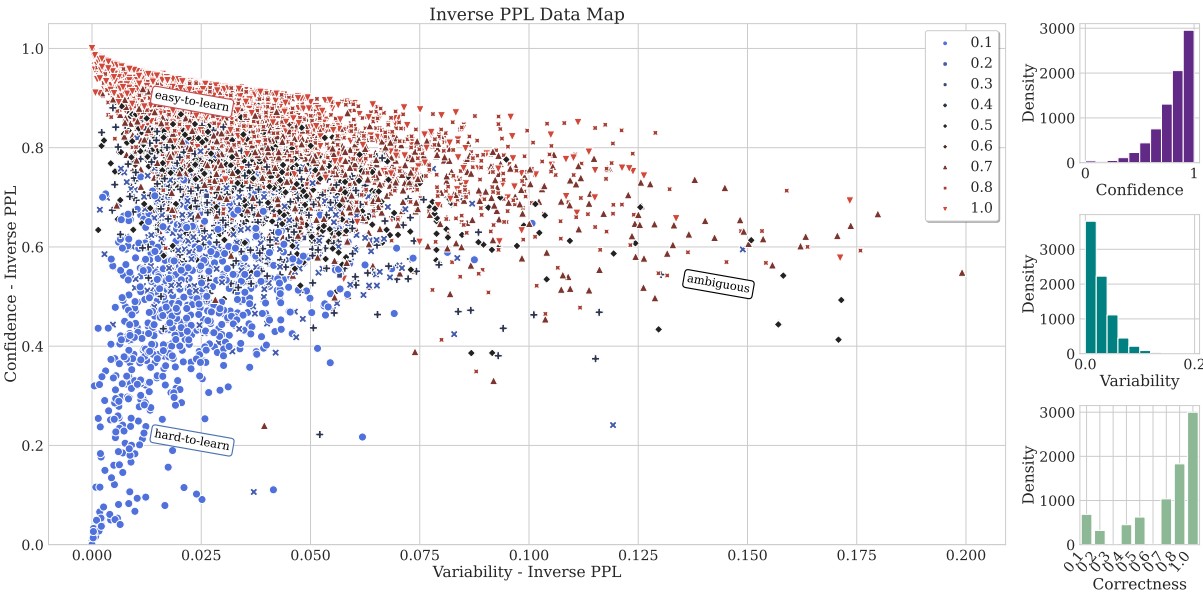

Figure 2: **Data map of COGS train set** for the Transformer model based on Inv PPL measure (converge epoch 10). The *x*-axis shows the **variability** and the *y*-axis the **confidence**. The colors and shapes indicate the **correctness**.

ing compositional generalization. In the first setup, we utilize subsets comprising 33% of the original datasets. These subsets are categorized as *hard-to-learn*, *ambiguous*, and *easy-to-learn* based on the extracted maps. For the second setup, we train models using subsets sized at 50% of the original datasets. Along with the *hard-to-learn*, *ambiguous*, and *easy-to-learn* subsets, we construct combined subsets that are also half the size of the original dataset by merging two 33% subsets selected based on the same confidence measure. Specifically, we select 33% of the examples from the more informative subset and allocate the remaining 17% from the other subset, following Swayamdipta et al. (2020). As will be discussed in the next section, our findings demonstrate that *hard-to-learn* samples have a more pronounced impact on model performance compared to *ambiguous* and *easy-to-learn* samples, and thus we consider them as more informative. When combining *ambiguous* and *easy-to-learn* samples, we consider including a greater number of *ambiguous* samples than *easy-to-learn* samples. If the union of these subsets is smaller than half of the whole training data, we randomly add samples to reach the 50% dataset size. Furthermore, we address the out-of-vocabulary (OOV) problem during subset selection by incorporating training samples from the entire dataset if they increase the vocabulary size. On the contrary, we remove the least informative samples that do not reduce the vocabulary size, ensuring consistent sub-

set sizes throughout the experiments. The statistics about the subsets obtained from the data maps are provided in Appendix A.4.

In addition to our subset selection experiments, we explore the potential of leveraging dataset cartography as a criterion for curriculum learning (CL). In particular, we adopt two CL approaches proposed by Hacohen and Weinshall (2019) and Zhang et al. (2019). We experiment with a fixed exponential pacing schedule using default hyperparameters in the former. We set the starting percentage to 4% and increase the scale to 1.9. On the other hand, the second CL method by Zhang et al. (2019) involves sorting examples based on a given criterion and dividing them into 10 equal-sized bins, resulting in a 10-stage curriculum. Within each bin, the examples are further sorted based on their lengths, and then each sorted bin is divided into non-overlapping batches. We distribute these batches randomly during training to avoid potential selection bias. Since we train our models for a fixed number of steps, after completing $1/10^{th}$ of the training, we unlock the second bin in a similar fashion.

### 3.4 Impact of Selected Subsets

We conduct a thorough analysis to understand the effect of subset selection on the training process and how the selection process impacts the subsequent generalization abilities of the models. Our key findings are summarized in Table 2 for the

|  |  | CFQ | | | COGS | | |
|---|---|---|---|---|---|---|---|
|  |  | **Inv PPL** | **CHIA** | **BLEU** | **Inv PPL** | **CHIA** | **BLEU** |
| 33% train | *easy-to-learn* | $12.19_{1.20}$ | $12.42_{0.59}$ | $9.88_{1.83}$ | $0.00_{0.00}$ | $0.06_{0.11}$ | $0.04_{0.07}$ |
|  | *ambiguous* | $17.69_{0.47}$ | $23.51_{0.86}$ | $20.99_{1.91}$ | $3.26_{5.61}$ | $20.30_{3.58}$ | $26.69_{4.17}$ |
|  | *hard-to-learn* | $\mathbf{36.55_{0.55}}$ | $\mathbf{34.98_{0.67}}$ | $\mathbf{34.71_{1.12}}$ | $\mathbf{53.50_{6.80}}$ | $\mathbf{45.41_{12.5}}$ | $\mathbf{50.56_{3.07}}$ |
|  | *random* | | $\underline{34.02_{1.09}}$ | | | $18.66_{6.72}$ | |
| 100% training | | | $38.71_{1.01}$ | | | $42.54_{7.62}$ | |

Table 2: Accuracy results for CFQ and COGS datasets. Models are trained on different 33% subsets of the train data compared to using the full dataset. The scores are averaged over 3 runs, where std. dev. is shown as a subscript. The best and second-best performing subsets are highlighted in bold and underlined, respectively. *Hard-to-learn* subset consistently performs better than the *random* subset, even outperforming 100% train set on the COGS dataset.

|  |  | CFQ | | | COGS | | |
|---|---|---|---|---|---|---|---|
|  |  | **Inv PPL** | **CHIA** | **BLEU** | **Inv PPL** | **CHIA** | **BLEU** |
| 50% train | *easy-to-learn* | 21.13 | 20.96 | 17.04 | 0.000 | 0.000 | 0.695 |
|  | *ambiguous* | 23.03 | 28.80 | 24.31 | 0.047 | 36.09 | 35.14 |
|  | *hard-to-learn* | **42.45** | 40.13 | 37.45 | **47.48** | **42.40** | **45.20** |
|  | *ambiguous + easy-to-learn* | 18.52 | 26.18 | 20.77 | 0.048 | 18.33 | 25.69 |
|  | *hard-to-learn + ambiguous* | 36.54 | 36.87 | 37.13 | 41.13 | 35.08 | 41.16 |
|  | *hard-to-learn + easy-to-learn* | 35.91 | **41.29** | **39.29** | 40.82 | 37.94 | 40.96 |
|  | *random* | | 35.16 | | | 30.24 | |
| 100% training | | | 37.71 | | | 36.80 | |

Table 3: Accuracy results for CFQ and COGS datasets. Models are trained on different 50% subsets of the train data compared to using the full dataset. The best and second-best performing subsets are highlighted in bold and underlined, respectively. It is worth mentioning that solely training on *hard-to-learn* samples or combining them with *easy-to-learn* samples outperforms using 100% training samples.

subsets comprising 33% of the original datasets. Our experimental results show that training models on *hard-to-learn* samples consistently yields superior generalization performance compared to training on ambiguous samples. Notably, the performance of *hard-to-learn* subsets surpasses that of *random* subsets overall, and for the COGS dataset, it even outperforms training on the entire training set. Training the models on *easy-to-learn* samples, on the other hand, leads to poor generalization performance. We also observe that Inverse Perplexity is a more effective measure than CHIA or BLEU for selecting samples based on their difficulty.

As we increase the subset size to 50% of the original dataset, our experimental results demonstrate significant improvements compared to full dataset training, as shown in Table 3. In the CFQ dataset, the accuracy of the *hard-to-learn (Inv PPL)* subset exceeds that of the full training by over 4%. When considering the CHIA measure, both the *hard-to-learn* and *hard-to-learn+easy-to-learn* subsets out-

perform 100% training. However, when using the BLEU measure, only the *hard-to-learn+easy-to-learn* subset surpasses the 100% training performance. Although the subset combinations show promising results with the CHIA and BLEU measures, they are still outperformed by the *hard-to-learn (Inv PPL)* subset. In COGS, we observe even more substantial improvements in accuracy. Across all measures, the *hard-to-learn* subset demonstrates an accuracy increase of over 5%, with the *hard-to-learn (Inv PPL)* subset outperforming the 100% training by over 10% accuracy. Notably, selecting 50% of the *hard-to-learn* samples consistently outperforms the subset combinations for all measures. While combining subsets does yield performance improvements in certain measures, it also highlights the limited effectiveness of these measures in effectively separating the different classes of instances. This is evident as the *hard-to-learn (Inv PPL)* subset consistently outperforms the subset combinations in both the CFQ and COGS datasets.

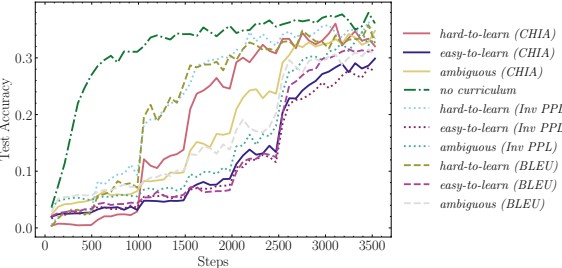

Figure 3: Accuracy plots on CFQ for the CL strategy by Hacohen and Weinshall (2019).

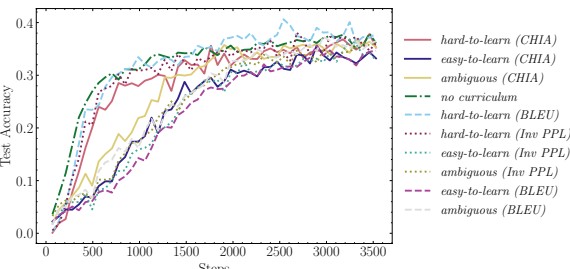

Figure 4: Accuracy plots on CFQ for the CL strategy by Zhang et al. (2019)

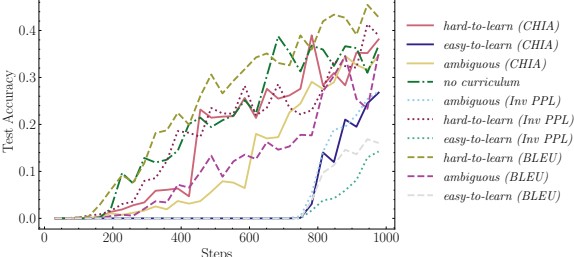

Figure 5: Accuracy plots on COGS for the CL strategy by Hacohen and Weinshall (2019).

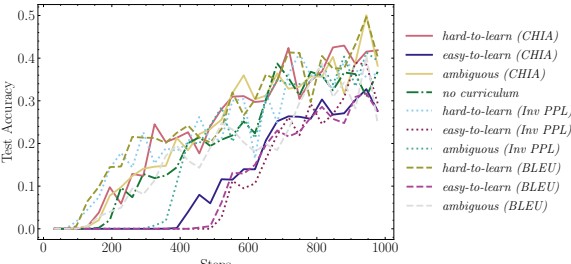

Figure 6: Accuracy plots on COGS for the CL strategy by Zhang et al. (2019).

## 3.5 Impact of Cartography-Based Curriculum Learning

We use dataset cartography to examine the impact of training dynamics on curriculum learning. Curriculum learning is a strategy that trains models on instances from easy to hard, based on the assumption that this order facilitates learning. However, we also explore the opposite strategy, which trains models on instances from hard to easy, and compare it with the conventional curriculum learning approach. This way, we can study how different training schedules affect the model performance.

Figure 3 depicts accuracy plots showing the performance of various CL strategies based on (Hacohen and Weinshall, 2019) on the CFQ dataset. The figure legends indicate the ranking scheme and the employed confidence measure. For instance, *hard-to-learn (Inv PPL)* refers to the case where Inv PPL is being used as the confidence measure, and the inclusion of the *hard-to-learn* samples is prioritized within the curriculum. Our analysis reveals that no single curriculum consistently outperforms others on the CFQ dataset. Exponential pacing leads to stagnant performance in the final $2/7^{th}$ of the training process due to surpassing the training size percentages of $33\%$ and $50\%$. Surprisingly, initiating training with *hard-to-learn* samples yields superior performance compared to *easy-to-learn* samples, contrary to common curriculum learn-

ing expectations. This aligns with our previous findings, emphasizing the benefits of starting with challenging examples for improved adaptation.

Figure 4 examines the impact of leveraging data maps within the CL strategy proposed by Zhang et al. (2019) for compositional generalization. The *hard-to-learn (BLEU)* configuration outperforms the *no curriculum* strategy, albeit with no notable improvement in convergence speed. This outcome mirrors our observations using the CL framework developed by Hacohen and Weinshall (2019), where initiating training with harder samples leads to better performance. However, the *ambiguous* configurations perform similarly to *no curriculum*, while the *easy-to-learn* configurations yield worse results than the *no curriculum* approach.

In Figures 5 and 6, we gain deeper insights into the contributions of dataset cartography. Overall, *hard-to-learn (BLEU)* emerges as the most effective configuration in the plots. Surprisingly, *ambiguous (Inv PPL)* performs as the second-best configuration in Figure 6, while *hard-to-learn (Inv PPL)* holds this position in Figure 5. The *no curriculum* approach ranks third and fourth in these respective plots. Furthermore, the *easy-to-learn* configurations demonstrate the poorest final performance across both curriculum learning frameworks.

Analyzing the accuracy plots of curriculum

learning, we observe that initiating training with easier examples and gradually progressing to more challenging instances does not lead to accelerated convergence or improved final model performance. On the other hand, the subset experiments presented in Tables 2 and 3 show that training models on hard-to-learn examples result in better model performance. Furthermore, the CL results highlight that starting the curriculum with *hard-to-learn* samples results in enhanced final performance. These findings, combined with the observation that the first unlocked examples are encountered more frequently during training, suggest that the superiority of *hard-to-learn* curricula over the *no curriculum* can be attributed to the increased exposure to challenging instances throughout the training process.

To sum up, our experimental findings highlight the effectiveness of utilizing dataset cartography for training subset selection and curriculum learning in the context of compositional generalization. Our results consistently show that leveraging dataset cartography leads to improved generalization performance. While curriculum learning also contributes to performance enhancement, its impact appears to be smaller compared to the use of dataset cartography for subset selection.

## 4   Related Work

Swayamdipta et al. (2020) use **training dynamics** to create data maps that categorize the dataset into three groups: easy-to-learn, hard-to-learn, and ambiguous. In a similar vein, Toneva et al. (2019) employ training dynamics for dataset categorization, specifically in classification tasks, by identifying misclassified or forgotten instances. On the contrary, the adversarial filtering algorithm proposed by Bras et al. (2020) ranks instances based on predictability, suggesting the removal of easy-to-learn examples. However, our research presents contrasting findings. Our experimental analyses show that combining the easy-to-learn category with other categories can improve the generalization performance. In another recent study, Wang et al. (2022) explore the relationship between the generalization performance and the training dynamics in an active learning setting. Their approach revolves around the adaptive selection of samples for labeling to obtain comparable or better performance with less training data. Notably, they discovered a robust correlation between the convergence speed of training and the resulting generalization performance. By

leveraging this connection, they propose a strategy to enhance overall generalization performance.

Contemporary research on **compositional generalization** focuses on two main aspects: proposing new datasets to explore model generalization capabilities and introducing novel techniques to address the compositionality problem.

As one of the early sample datasets, SCAN (Lake and Baroni, 2017) simulates a navigation task, and measures generalization to longer splits or systematic generalization with new verbs in different splits. Keysers et al. (2020) define a mathematically rigorous way to create compositional datasets and create CFQ dataset which is a semantic parsing task of generating SPARQL queries from natural language questions. Kim and Linzen (2020) proposed COGS for semantic parsing, where the source is an English string, and the target is its logical form.

In terms of novel techniques, researchers propose various approaches for compositional generalization. These include creating novel architectures to solve compositionality problem (Perez et al., 2018; Hudson and Manning, 2018), modifying existing architectures for better generalization (Russin et al., 2020; Akyurek and Andreas, 2021), utilizing different learning paradigms such as meta-learning (Lake, 2019) and pre-training (Furrer et al., 2020), or data augmentation (Andreas, 2020; Qiu et al., 2022). With the rise of large language models (LLMs), choosing in-context examples for better compositional generalization with LLMs (Levy et al., 2022; An et al., 2023) is another open research problem. In the compositional generalization literature, only a few studies investigated the impact of training dynamics on generalization performance. For instance, studies by both Liu et al. (2020) and Chen et al. (2020) propose curriculum learning schemes to help models learn accurate execution traces for lengthy training samples. They divide the samples into partitions based on the length and train the models sequentially, starting with the shortest examples. In contrast, our work takes a different approach by utilizing training dynamics to create data maps and leveraging them for compositional generalization. This is achieved either through subset selection or curriculum criteria.

## 5   Conclusion

Transformers are great at language modeling and various downstream tasks, but their ability to

achieve compositional generalization compared to humans remains debatable. In this study, we addressed this challenge by demonstrating that selecting a subset of the training dataset using dataset cartography and training models on this subset can enhance model accuracy by up to $10\%$. We showed that our setup can generalize to different model architectures and natural datasets. Moreover, we achieved improved performance by employing a dataset cartography-based curriculum learning without the need for hyperparameter tuning. Looking ahead, we anticipate that this research direction promises insights into the necessary syntax, semantics, and structure for informative data instances, informing the development of novel data augmentation strategies and advancing our understanding of deep models' generalization capabilities.

## Limitations

**Synthetic data.** While not a limitation specific to our approach, many of the datasets used for compositional generalization, including CFQ (Keysers et al., 2020) and COGS (Kim and Linzen, 2020), are synthetically generated. Hence, they may not cover all the complexities of natural languages. While we perform initial experiments with SMCalFlow-CS Simple (Meron, 2022), a natural compositional generalization dataset, additional experiments are necessary to make conclusive statements.

**Other models.** In our work, we performed experiments with vanilla Transformers (Vaswani et al., 2017) and Bi-LSTM with attention (Bahdanau et al., 2015). While we conduct preliminary experiments on Bi-LSTM with attention and demonstrate that these results are in line with primary experiments, more experiments are needed to make conclusive remarks about other neural models.

## Ethics Statement

This work consists of running a lot of experiments on GPU clusters, therefore it requires energy and increases carbon footprint. For this reason, best practices to protect the environment are employed. All datasets and used code are utilized with permissive licenses and will be made available.

## Acknowledgment

This work was partly supported by the KUIS AI Center Fellowship to Osman Batur İnce.

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

## A Supplementary

### A.1 Reproducibility

We use the experimental setup created in Csordás et al. (2021) for vanilla Transformers and modify it to calculate and store training dynamics, and implement a curriculum learning framework. After storing training dynamics such as perplexity, CHIA, and BLEU, we choose a subset with a criteria. As we choose random subsets as a baseline, we specify the seed during this process. For the Bi-LSTM with attention experiments, we adopt the setup created in Patel et al. (2022).

As mentioned in the paper, we used models and hyperparameters from Csordás et al. (2021) to ease computational constraints and use a initial strong random baseline (see Table 4). Accuracy is calculated on the sequence level, meaning that all tokens in the output sequence should match all tokens in the gold sequence while preserving the sequence. We use the NLTK BLEU-4 score as the BLEU metric. We specifically use `SmoothingFunction.method4` for smoothing and `auto_reweigh` is set to `True` as some examples are shorter than 4 words. For the CFQ dataset, we applied the same preprocessing as in Csordás et al. (2021), which is used in Keysers et al. (2020) as well. For COGS, no dataset preprocessing was used.

### A.2 Variability Equations

Equations for variability calculations, when CHIA, inverse perplexity, or BLEU measures are used, are shown from equations 6 to 8 respectively.

$$v_i = \sqrt{\frac{\sum_{e=1}^{E}\left(\frac{1}{T}\sum_{t=1}^{T} p_{\theta^e}\left(y_{it}^* \mid \mathbf{x}_i\right) - \hat{\mu}_i\right)^2}{E}} \tag{6}$$

$$v_i = \sqrt{\frac{\sum_{e=1}^{E}\left(\prod_{t=1}^{T} \sqrt[T]{p_{\theta^e}\left(y_{it}^* \mid \mathbf{x}_i\right)} - \hat{\mu}_i\right)^2}{E}} \tag{7}$$

$$v_i = \sqrt{\frac{\sum_{e=1}^{E}\left(\text{BLEU}(\hat{\mathbf{y}}_{\mathbf{i}}^{(\mathbf{e})}, \mathbf{y}_i) - \hat{\mu}_i\right)^2}{E}} \tag{8}$$

where $\hat{\mu}_i$ (confidence) values are calculated as shown in the main paper (see Equations 3–5, re-

spectively). In these equations, $i$ denotes the instance, $E$ represents the total number of epochs, $\mathbf{x}_i$ is the input sequence, $\theta^{(e)}$ corresponds to the set of model parameters at epoch $e$. Additionally, $y_{i_t}^*$ corresponds to the $i$-th token of the ground truth output sequence $\mathbf{y}_i$ of length $T$, $\hat{\mathbf{y}}_{\mathbf{i}}^{(e)}$ refers to the predicted sequence generated by the model parameters at epoch $e$.

### A.3 Additional Experiments

While our primary experiments focus on training Transformers on the CFQ and COGS datasets, we conduct supplementary experiments with Bi-LSTM with attention model on the COGS task and Transformer model on two SMCS splits. This approach allows us to examine the transferability of our results from synthetic datasets and the Transformer to other architectures and natural datasets.

For the SMCS 16 and 32 splits, we see a similar trend compared to CFQ and COGS results (see Table 5 and 3). The performance of *hard-to-learn* subsets exceeds the original dataset performance consistently. Even if *hard-to-learn* examples consisted of manual annotation errors in Swayamdipta et al. (2020), training models on *hard-to-learn* subsets improve performance for the SMCS splits. However, this finding does not necessarily translate to training models on errors result in better performance. Rather, the contribution of *hard-to-learn* subsets significantly obscures any performance degradation that may occur from annotation errors. Similarly, *easy-to-learn* subsets perform the worst among all of the subsets. Although *hard-to-learn* subsets perform the best among the other subsets, the ranking between metrics is more fluid compared to the CFQ and COGS results. While *hard-to-learn (Inv PPL)* outperform other subsets persistently in the CFQ and COGS results (Table 3), *hard-to-learn (BLEU)* and *hard-to-learn (CHIA)* are the best performing subsets in SMCS 16 and 32 subsets respectively.

Table 6 shows the Bi-LSTM with attention performance on the COGS dataset. Surprisingly, the Bi-LSTM performance is much worse than the Transformer performance. Nonetheless, these results are consistent with the results in the original COGS dataset (Kim and Linzen, 2020).

Similar to the SMCS experiments, the Bi-LSTM experiments support our primary findings. Training models on *hard-to-learn* subsets continuously outperform training on the full dataset. Compared to

| Dataset | $d_{model}$ | $d_{ff}$ | $n_{head}$ | $n_{layer}$ | batch size | learning rate | warmup | scheduler | $n_{param}$ |
|---------|-------------|----------|------------|-------------|------------|---------------|--------|-----------|-------------|
| CFQ | 128 | 256 | 16 | 2 | 1024 | 0.9 | 4000 | Noam | 685k |
| COGS | 512 | 512 | 4 | 2 | 128 | $10^{-4}$ | - | - | 9.3M |

Table 4: Hyperparameters and number of parameters for each task. Feedforward size is denoted as $d_{ff}$. Only CFQ batch size is changed from Csordás et al. (2021) (4096 → 1024).

| | SMCS 16 | | | SMCS 32 | | |
|---|---|---|---|---|---|---|
| | Inv PPL | CHIA | BLEU | Inv PPL | CHIA | BLEU |
| *easy-to-learn* | 0.0 | 0.0 | 0.0 | 0.1 | 0.0 | 1.5 |
| *ambiguous* | 0.0 | 2.0 | 2.3 | 7.0 | 7.8 | 11.6 |
| *hard-to-learn* | **4.5** | **4.5** | **6.8** | **16.8** | **17.5** | **15.9** |
| *random* | 0.4 | 0.4 | 0.4 | 5.9 | 5.9 | 5.9 |
| 100% train | | 4.2 | | | 15.6 | |

Table 5: Accuracy results for the SMCS 16 and 32 splits. Models are trained on different 50% subsets of the train data instead of the full train set. The best performing subset is given in bold. Training models only on *hard-to-learn* samples outperforms using 100% train data.

| | Inv PPL | CHIA | BLEU |
|---|---|---|---|
| *easy-to-learn* | 2.7 | 8.3 | 9.9 |
| *ambiguous* | 14.2 | 2.8 | 11.9 |
| *hard-to-learn* | **16.6** | **15.7** | **20.2** |
| *random* | | 9.9 | |
| 100% train | | 13.7 | |

Table 6: Accuracy results for the COGS dataset. Models are trained on different 50% subsets of the train data instead of the full train set. The best performing subset is given in bold. Training models solely on *hard-to-learn* samples outperforms using 100% train data.

| | | Length | Word rarity |
|---|---|---|---|
| | *random* | 13.56 / 27.78 | 3.97 / 3.47 |
| **Inv PPL** | *easy-to-learn* | 11.86 / 24.55 | 3.99 / 3.41 |
| | *ambiguous* | 11.51 / 23.64 | 4.06 / 3.51 |
| | *hard-to-learn* | 15.82 / 32.13 | 3.95 / 3.54 |
| **CHIA** | *easy-to-learn* | 10.91 / 22.34 | 4.02 / 3.43 |
| | *ambiguous* | 13.07 / 27.86 | 3.94 / 3.50 |
| | *hard-to-learn* | 16.40 / 33.94 | 3.93 / 3.53 |
| **BLEU** | *easy-to-learn* | 11.41 / 23.18 | 4.02 / 3.44 |
| | *ambiguous* | 12.78 / 25.11 | 3.96 / 3.47 |
| | *hard-to-learn* | 16.23 / 33.49 | 3.94 / 3.52 |

Table 7: Statistics about the subsets of the CFQ dataset on 33% selected instances based on Inv PPL, CHIA and BLEU measures. We report average input/output length and word rarity. Statistics are averaged over 3 runs.

the Transformer performance (Table 3), the maximum absolute performance increase between *hard-to-learn* subsets and full training decreases by 4%. However, the maximum relative performance increase between *hard-to-learn* subsets and full training increases, showing that dataset cartography improves generalization performance in architectures other than Transformer.

### A.4 Subsets Obtained from Data Maps

We examine four key statistics to gain insights into the nature of the subsets created through data cartography: (1) input length, (2) output length, (3) input word rarity, and (4) output word rarity. Word rarity is calculated as the sum of negative log word frequencies normalized with sentence length, as shown in Equation 9. In this equation, $T$ is the sequence length, $y_{it}^*$ is the $t^{th}$ gold token

for sequence $i$, and $f(y_{it}^*)$ denotes frequency of gold token $y_{it}^*$. Table 7 presents these statistics for the CFQ dataset, and reveals interesting patterns. Among these different subsets, the *hard-to-learn* subsets show longer input and output lengths compared to all other splits. Conversely, both *ambiguous* and *easy-to-learn* samples tend to be shorter in length compared to the *randomly selected* samples. Analyzing word rarities, we observe that the *hard-to-learn* subsets have lower input rarity but higher output rarity compared to the *random* subset. On the other hand, the *easy-to-learn* and *ambiguous* samples show higher input rarity than the *random* subset. Notably, the word rarity in *ambiguous* samples surpasses even that of the *hard-to-learn* samples. These statistics provide valuable insights into the subsets. However, determining whether dataset cartography is solely driven by factors such as length and rarity or represents a more complex distribution of samples remains a topic for future investigation.

The statistics of the COGS subsets, as presented in Table 8, show similar patterns to the CFQ subsets discussed in Table 7. Specifically, we observe that the *hard-to-learn* subsets tend to have

|  |  | **Length** | **Word rarity** |
|---|---|---|---|
|  | *random* | 7.47 / 43.52 | 4.54 / 3.34 |
| **Inv PPL** | *easy-to-learn* | 6.59 / 34.22 | 4.25 / 3.24 |
|  | *ambiguous* | 7.16 / 42.23 | 4.82 / 3.41 |
|  | *hard-to-learn* | 8.83 / 56.62 | 4.87 / 3.47 |
| **CHIA** | *easy-to-learn* | 6.61 / 33.78 | 4.24 / 3.23 |
|  | *ambiguous* | 7.81 / 48.53 | 4.90 / 3.46 |
|  | *hard-to-learn* | 8.83 / 56.87 | 4.87 / 3.48 |
| **BLEU** | *easy-to-learn* | 6.27 / 32.24 | 4.33 / 3.27 |
|  | *ambiguous* | 8.67 / 54.92 | 4.78 / 3.43 |
|  | *hard-to-learn* | 9.08 / 58.20 | 4.77 / 3.45 |

Table 8: Statistics about the subsets of the COGS dataset on 33% selected instances based on Inv PPL, CHIA, and BLEU measures. We report average input/output length and word rarity. Statistics are averaged over 3 runs.

longer samples compared to the *ambiguous* subsets, while the *ambiguous* subsets are longer than the *easy-to-learn* subsets. However, unlike the CFQ subsets, the COGS subsets display an interesting characteristic: as the subsets become harder, there is an increase in the presence of rare words both within and outside the dataset vocabulary. This phenomenon can be attributed to the larger vocabulary size and smaller dataset size of the COGS dataset, as outlined in Table 1. Consequently, the variability in word usage plays a more prominent role in determining the hardness of the data instances in COGS. Therefore, by employing dataset cartography, we are able to select subsets that exhibit different underlying factors, ultimately leading to dataset-specific improvements in performance.

$$\text{Rarity}(i) = -\frac{1}{T}\sum_{t=1}^{T}\log f\left(y_{it}^{*}\right) \qquad (9)$$

### A.5 Subset Examples

In the following, we randomly sample examples from 5% hardest-to-learn, most ambiguous, or easiest-to-learn examples. We show examples based on BLEU measure for the CFQ dataset (examples (1), (2), and (3)), and based on Inv PPL measure for the COGS dataset (examples (4), (5), and (6)), for brevity. While these examples are too small for any inference, we see that these examples reflect subset statistics mentioned in Tables 7 and 8.

**CFQ Subset Samples:**

(1) *An easy-to-learn sample:*

What did a child of M0 executive produce, edit, write, direct, and produce →

```
SELECT DISTINCT ?x0 WHERE { ?x0
film.film.directed_by ?x1 . ?x0
film.film.edited_by ?x1 . ?x0
film.film.executive_produced_by
?x1 . ?x0 film.film.produced_by|↩
ns:film.film.production_companies
?x1 . ?x0 film.film.written_by ?x1
. ?x1 people.person.parents|↩
ns:fictional_universe.↩
fictional_character.parents|↩
ns:organization.organization.↩
parent/ns:organization.↩
organization_relationship.parent
M0 }
```

(2) *An ambiguous sample:*

What did M0 found and M1's female founder found →

```
SELECT DISTINCT ?x0 WHERE { ?x0
organization.organization.founders
?x1 . ?x0
organization.organization.founders
M0 . ?x1 organization.↩
organization_founder.↩
organizations_founded M1 . ?x1
people.person.gender m_02zsn }
```

(3) *A hard-to-learn sample:*

Was M2 a film producer that employed a spouse of M1, employed M0's executive producer, and employed M4 →

```
SELECT count (*) WHERE { ?x0
film.producer.↩
films_executive_produced M0 . ?x1
people.person.spouse_s/↩
ns:people.marriage.spouse|↩
ns:fictional_universe.↩
fictional_character.married_to/↩
ns:fictional_universe.↩
marriage_of_fictional_↩
characters.spouses M1 . FILTER (
?x1 != M1 ) . M2 a film.producer .
M2 business.employer.employees/↩
ns:business.employment_tenure.↩
person ?x0 . M2
business.employer.employees/↩
ns:business.employment_tenure.↩
```

```
person ?x1 . M2
business.employer.employees/↵
ns:business.employment_tenure.↵
person M4 }
```

**COGS Subset Samples:**

(4)  *An easy-to-learn sample:*

A cake was drawn by Emma . →

```
cake ( x _ 1 ) AND draw . theme (
x _ 3 , x _ 1 ) AND draw . agent (
x _ 3 , Emma )
```

(5)  *An ambiguous sample:*

The cat wished to sleep . →

```
* cat ( x _ 1 ) ; wish . agent (
x _ 2 , x _ 1 ) AND wish . xcomp (
x _ 2 , x _ 4 ) AND sleep . agent
( x _ 4 , x _ 1 )
```

(6)  *A hard-to-learn sample:*

James gave a lion a cake in the fridge . →

```
* fridge ( x _ 8 ) ; give . agent
( x _ 1 , James ) AND give .
recipient ( x _ 1 , x _ 3 ) AND
give . theme ( x _ 1 , x _ 5 ) AND
lion ( x _ 3 ) AND cake ( x _ 5 )
AND cake . nmod . in ( x _ 5 ,
x _ 8 )
```

### A.6  Detailed Error Analysis

To gain further insights into the performance of our model, we conduct a comprehensive manual error analysis on both the CFQ and COGS datasets. Our objective was to identify the specific test samples where the model exhibits improved performance after training with a selected subset, and to determine the general properties of these samples.

**On the CFQ dataset.** Our analysis reveals that the *hard-to-learn (Inv PPL)* model outperforms the 100% trained model, particularly on sentences that are shorter than average or of average length. This observation highlights the effectiveness of dataset cartography in enhancing compositional generalization, without relying on spurious correlations. However, it is important to note that the *hard-to-learn (Inv PPL)* model does not demonstrate the same level of improvement in generalizing to longer sentences. Figure 7 provides further insights into the performance of the models. We observe a slight

increase in errors for the shortest samples. This can be attributed to the fact that the *hard-to-learn (Inv PPL)* model generates longer outputs than the target for a subset of these samples ((7)). This behavior can be explained by the length bias present in the *hard-to-learn* subsets, as the models tend to generate longer outputs when encountering instances longer than the average length.

(7)  Was a film director M0 →

GOLD: SELECT count ( * ) WHERE {
M0 a film.director }

OUT: SELECT count ( * ) WHERE { M0
a film.director . M0
film.director.film M2 }

Errors observed in both models exhibit a systematic nature. For instance, there are samples where the models fail to correctly order the triple sequences, resulting in incorrect output (see Example (8)). Another common error type involves swapping the 1st argument in a triple with the 3rd argument ((9), formatted for spacing). It is worth noting that these error patterns are not specific to either the *hard-to-learn* or the 100% trained models.

(8)  Was M0 ' s prequel a film →

GOLD: SELECT count ( * ) WHERE {
?x0 a film.film . ?x0
film.film.sequel M0  }

OUT: SELECT count ( * ) WHERE {
?x0 film.film.sequel M0 . ?x0 a
film.film }

(9)  Was a character M1 ' s director →

GOLD: SELECT count(*) WHERE { ?x0
a fictional_universe.↵
fictional_character . ?x0
film.director.film M1 }

OUT: SELECT count(*) WHERE { ?x0 a
fictional_universe.↵
fictional_character . M1
film.director.film ?x0 }

**On the COGS dataset.** In the COGS dataset, each instance belongs to one of the 21 generalization categories, such as *Passive → Active*, where the verb structure is transformed from a passive form to an active form (e.g. "The book was **squeezed**". → "The girl **squeezed** the strawberry."). This cate-

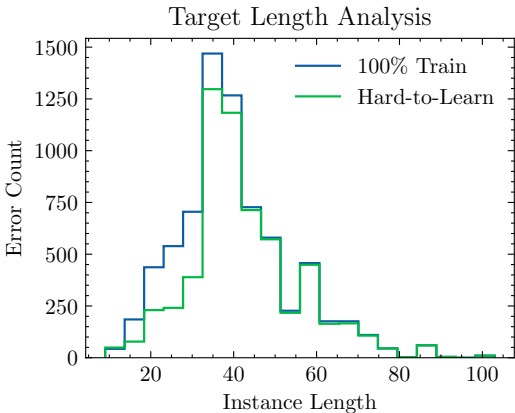

Figure 7: Target length histogram for test errors on CFQ.

gorization allows us to explore the changes in accuracies across different categories when comparing models trained on 33% *hard-to-learn* subsets and the 100% training dataset (refer to Table 10).

For *Inv PPL*, we see performance increase nearly in all of the categories compared to full training except 7 categories. In 3 of these categories, all 4 models have 0% accuracy. And from the remaining 4 categories, in only one category the performance discrepancy is remarkable (*Subject → Object (common noun)*). While dataset cartography significantly contributes towards lexical generalization, the contribution towards structural generalization remains limited.

Among models trained with subsets, *Inv PPL* outperforms *CHIA* on almost all categories, and while overall *Inv PPL* performs better, the ranking between *Inv PPL* and *BLEU* is more volatile. Results in Table 10 indicate that the Inv PPL measure can distinguish informative examples better compared to the CHIA measure. Moreover, different confidence measures have different characteristics, therefore they can give more importance to measure-specific instances.

### A.7 Remaining Cartography Plots

We present the remaining cartography plots for the CFQ and COGS datasets in this section. Same as previous plots, we only plot randomly sampled 33% of the training set. For the CFQ dataset, these remaining plots include the Inv PPL plot (Figure 8) and the CHIA plot (Figure 9). Similarly, for the COGS dataset, the remaining plots consist of the BLEU plot (Figure 10) and the CHIA plot (Figure 11).

Upon examining these plots, we observe distinct characteristics among them. The CHIA plots ap-

| Motivation | | | |
|---|---|---|---|
| *Practical* □ △ ◯ | *Cognitive* | *Intrinsic* □ △ ◯ | *Fairness* |
| **Generalisation type** | | | | | |
| *Compositional* □ △ ◯ | *Structural* | *Cross Task* | *Cross Language* | *Cross Domain* | *Robustness* |
| **Shift type** | | | |
| *Covariate* □ △ ◯ | *Label* | *Full* | *Assumed* |
| **Shift source** | | | |
| *Naturally occuring* | *Partitioned natural* ◯ | *Generated shift* | *Fully generated* □ △ |
| **Shift locus** | | | |
| *Train–test* □ △ ◯ | *Finetune train–test* | *Pretrain–train* | *Pretrain–test* |

Table 9: GenBench eval card (Hupkes et al., 2022) of our work

pear denser, with data examples concentrated in specific regions. In contrast, the BLEU plots exhibit a more widespread distribution, while the Inv PPL plots demonstrate the highest degree of dispersion. These plots offer interesting insights when comparing the performances of CHIA and Inv PPL measures. As instances in Inv PPL plots are better distributed compared to instances in CHIA plots, categories of examples are more distinguishable, resulting in CHIA *hard-to-learn* subsets including *ambiguous* or even *easy-to-learn* instances. Despite their similar underlying mathematical intuition, these plots contribute to a better understanding of the observed differences.

### A.8 GenBench Eval Card

We provide the GenBench eval card (Table 9) to help centralize the generalization evaluation in state-of-the-art language models. We conduct many experiments in our paper, but we can safely divide them into 3 main categories by their datasets (CFQ, COGS, and SMCS).

The motivation beyond our experiments is the same. We hypothesized that we could improve the compositional generalization performance of neural models by harnessing their training dynamics to construct a smaller training set. Therefore, our motivation is practical as we aim to achieve better generalization and intrinsic as we examine and utilize models' training dynamics at the same time. All of our datasets are compositional gener-

| Category | hard-to-learn (33% train) | | | 100% training |
|---|---|---|---|---|
| | Inv PPL | CHIA | BLEU | |
| Subject → Object (common noun) | 0.45 | 0.57 | 0.85 | 0.61 |
| Subject → Object (proper noun) | 0.08 | 0.05 | 0.16 | 0.08 |
| Object → Subject (common noun) | 0.95 | 0.95 | 0.97 | 0.90 |
| Object → Subject (proper noun) | 0.54 | 0.31 | 0.40 | 0.41 |
| Primitive noun → Subject (common noun) | 0.93 | 0.58 | 0.92 | 0.63 |
| Primitive noun → Subject (proper noun) | 0.90 | 0.73 | 0.81 | 0.44 |
| Primitive noun → Object (common noun) | 0.47 | 0.19 | 0.14 | 0.15 |
| Primitive noun → Object (proper noun) | 0.19 | 0.27 | 0.15 | 0.14 |
| Primitive verb → Infinitival argument | 0.48 | 0.37 | 0.07 | 0.26 |
| Object-modifying PP → Subject-modifying PP | 0.00 | 0.00 | 0.00 | 0.00 |
| Depth generalization: Sentential complements | 0.00 | 0.00 | 0.00 | 0.00 |
| Depth generalization: PP modifiers | 0.09 | 0.07 | 0.08 | 0.07 |
| Active → Passive | 0.98 | 0.94 | 0.89 | 0.99 |
| Passive → Active | 0.72 | 0.57 | 0.74 | 0.40 |
| Object-omitted transitive → Transitive | 0.89 | 0.65 | 0.81 | 0.87 |
| Unaccusative → Transitive | 0.46 | 0.45 | 0.49 | 0.41 |
| Double object dative → PP dative | 0.55 | 0.48 | 0.71 | 0.54 |
| PP dative → Double object dative | 0.65 | 0.42 | 0.43 | 0.17 |
| Agent NP → Unaccusative Subject | 0.45 | 0.48 | 0.39 | 0.29 |
| Theme NP → Object-omitted transitive Subject | 0.74 | 0.74 | 0.85 | 0.80 |
| Theme NP → Unergative subject | 0.71 | 0.71 | 0.77 | 0.78 |

Table 10: COGS accuracy by generalization categories. Subsets have 33% of the original dataset size. Each result is averaged over 3 runs.

alization datasets and they display covariate shift. While SMCS is a partitioned neutral dataset, CFQ and COGS datasets are fully generated. All of our shift loci are train–test as we train Transformer and Bi-LSTM models from scratch.

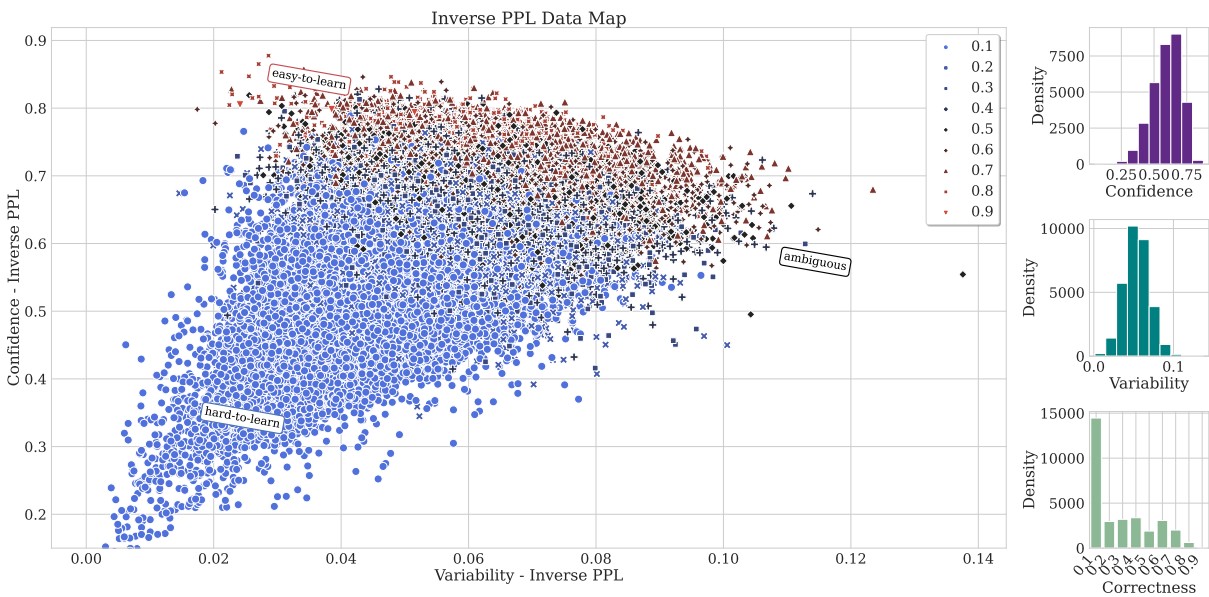

Figure 8: **Data map of CFQ train set** for the Transformer model based on Inv PPL measure (converge epoch 20). The *x*-axis shows the **variability** and the *y*-axis the **confidence**. The colors and shapes indicate the **correctness**.

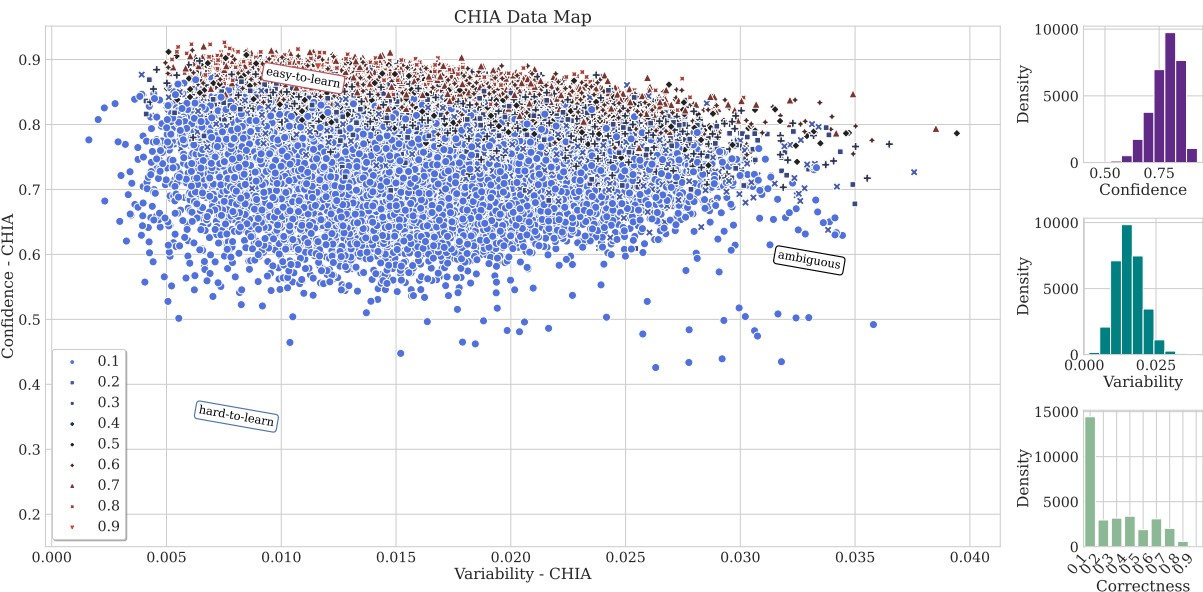

Figure 9: **Data map of CFQ train set** for the Transformer model based on CHIA measure (converge epoch 20). The *x*-axis shows the **variability** and the *y*-axis the **confidence**. The colors and shapes indicate the **correctness**.

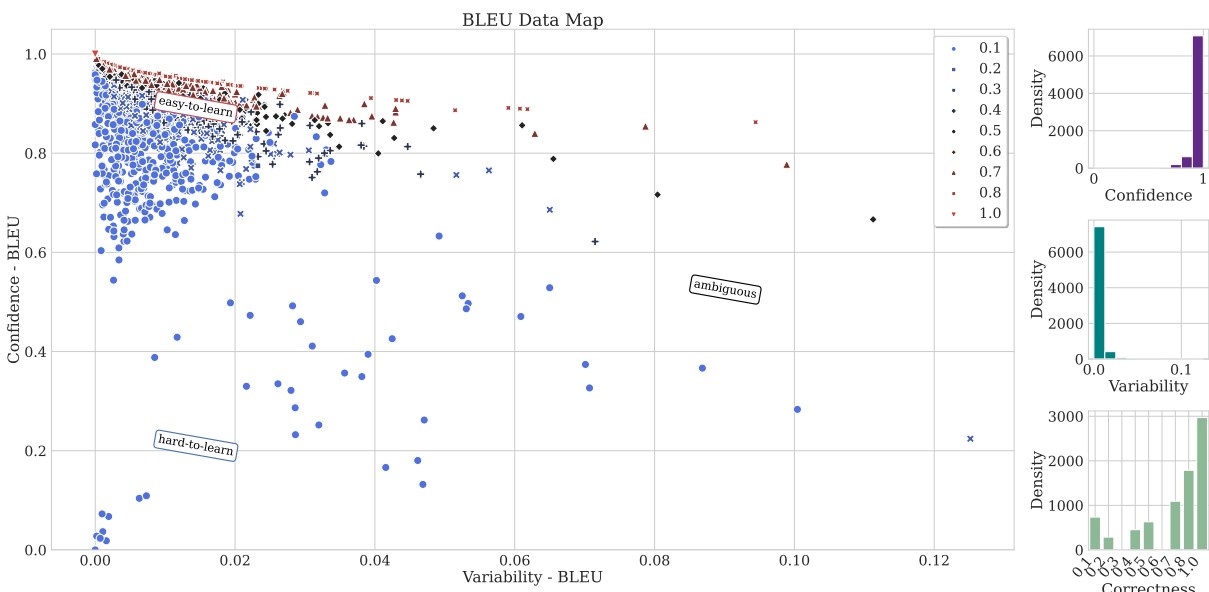

Figure 10: **Data map of COGS train set** for the Transformer model based on BLEU measure (converge epoch 10). The *x*-axis shows the **variability** and the *y*-axis the **confidence**. The colors and shapes indicate the **correctness**.

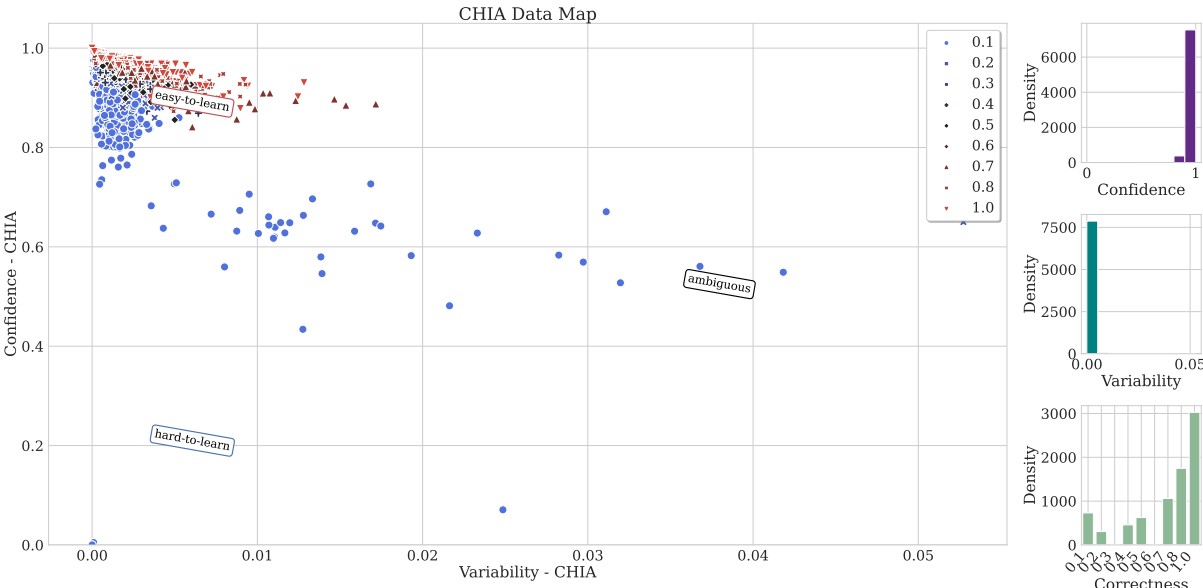

Figure 11: **Data map of COGS train set** for the Transformer model based on CHIA measure (converge epoch 10). The *x*-axis shows the **variability** and the *y*-axis the **confidence**. The colors and shapes indicate the **correctness**.