# OpenReview forum: "Harnessing Dataset Cartography for Improved Compositional Generalization in Transformers"
_EMNLP/2023/Conference — EMNLP 2023 Findings_

### Official Review · Reviewer_UCsN · 2023-07-31

**Soundness:** 4

**Excitement:**

4: Strong: This paper deepens the understanding of some phenomenon or lowers the barriers to an existing research direction.

**Paper Topic And Main Contributions:**

This paper introduces a method that leverages dataset cartography to improve compositional generalization in Transformer models. By strategically selecting relevant data (easy-to-learn, hard-to-learn etc.), the approach achieves up to 10% accuracy improvement on CFQ and COGS datasets. The findings highlight the potential of dataset cartography in enhancing the model's ability to understand and generate language more effectively.

**Questions For The Authors:**

- Have you tried out more complex model structures instead of the vanilla transformer architecture? If so, does that change your findings?

**Reasons To Accept:**

- Generally well written
- Well described and conducetd experiments
- Interesting results from which the community could benefit

**Reasons To Reject:**

- The paper yould greatly benefit from a wider range of compared model architectures

**Reproducibility:**

3: Could reproduce the results with some difficulty. The settings of parameters are underspecified or subjectively determined; the training/evaluation data are not widely available.

**Reviewer Confidence:**

2: Willing to defend my evaluation, but it is fairly likely that I missed some details, didn't understand some central points, or can't be sure about the novelty of the work.

**Typos Grammar Style And Presentation Improvements:**

- Figures: Please increase the size of the legends

---

> ### Author Rebuttal · Authors · 2023-08-28
>
> We sincerely thank the reviewer for their invaluable comments, insightful recommendations, and constructive critiques. We deeply appreciate that they find our paper well-written and our results interesting, with well-described and executed experiments.
>
> In response to the provided comments and questions:
>
> > The paper yould greatly benefit from a wider range of compared model architectures
>
> Recognizing the significance of this suggestion, we have expanded our model architecture analysis to include the Bi-LSTM architecture.  It's important to note that due to time constraints, we were only able to conduct experiments with the Bi-LSTM model on the COGS dataset. The outcomes of these additional experiments are presented in the following tables:
>
> __COGS - Bi-LSTM (50% train set)__
> | | InvPPL | CHIA | BLEU |
> |-------|----------:|-------------:|------:|
> |easy-to-learn | 2.7% | 8.3% | 9.9% |
> |ambiguous | 14.2% | 2.8% | 11.9% |
> |hard-to-learn | **16.6%** | **15.7%** | **20.2%** |
> |random | 9.9% | 9.9% | 9.9% |
> |100% train set | 13.7% | 13.7% | 13.7% |
>
> These supplementary experiments strongly indicate that dataset cartography offers significant enhancements in compositional generalization performance across various neural architectures. Notably, the insights drawn from these Bi-LSTM experiments are in alignment with the conclusions outlined in our paper for the Transformer architecture.
>
> > Have you tried out more complex model structures instead of the vanilla transformer architecture? If so, does that change your findings?
>
> Our experimentation has been currently limited to two neural architectures, namely the vanilla Transformer model and the newly added Bi-LSTM model. Although we haven't investigated more complex neural architectures at this stage, our empirical findings strongly suggest that the concept of dataset cartography holds potential for enhancing compositional generalization across a broader spectrum of neural architectures. Although we haven't ventured into investigating more complex structures at this stage, we will certainly highlight this perspective in our paper to provide a clearer insight into our research direction.
>
> Regarding the presentation improvements:
>
> > Figures: Please increase the size of the legends
>
> We increased the size of the legends to improve readability and clarity. Your feedback is greatly appreciated, and we ensured that the figure legends are more accessible in the revised version of the paper.
>
> To summarize, we genuinely thank the reviewer for their invaluable input, which significantly enriched our paper and strengthened our claims. While we have not tried more complex neural architectures, our empirical results for the vanilla Transformer and newly-added Bi-LSTM model strongly suggest that dataset cartography is applicable to a broader range of neural architectures. We certainly will highlight highlight this perspective to provide a clearer insight to our research direction.

---

### Official Review · Reviewer_xSzh · 2023-08-07

**Typos Grammar Style And Presentation Improvements:** NA
**Soundness:** 2

**Excitement:**

3: Ambivalent: It has merits (e.g., it reports state-of-the-art results, the idea is nice), but there are key weaknesses (e.g., it describes incremental work), and it can significantly benefit from another round of revision. However, I won't object to accepting it if my co-reviewers champion it.

**Missing References:**

NA

**Paper Topic And Main Contributions:**

Dataset cartography is technique that quantifies the variability and confidence associated with each training sample.


The paper investigates use dataset cartography to identify a subset of compositional generalization data. The authors further implement dataset cartography as a criterion for curriculum learning, mitigating the need for hyperparameter tuning while delivering superior performance. The enhancement is demonstrated through a 10% increase in accuracy on CFQ and COGS datasets.

**Questions For The Authors:**

1. Why the cartography + curriculum learning could improve compositional generalization? What if we simply train the model with enough time and regularization? What if we use some simple metric to identify the hard example (e.g., training acc) combined with curriculum learning?

**Reasons To Accept:**

Compositional Generalization is a key problem that existing LM lacks, and this paper uses dataset cartography to identify those hard examples to improve such accuracy (via subset sampling containing hard examples) with curriculum learning



(post-rebuttal)

After reading the rebuttal, I agree that the finding that hard samples found via cartography can help compositional generalization is interesting, so I'd like to increase the excitement to "Ambivalent".

**Reasons To Reject:**

The dataset cartography used in this paper seem to be another way to identify the hard examples, and the way authors utilize it is mainly about curriculum learning (easti-to-hard). I didn't fully get how this is related to compositional generalization. The composiontality is not easily shown in the cartography, and we could not simply assume hard examples are those composite data. Please correct me if I'm wrong, but I think authors need to provide more justification how the proposed method could enable models for better "compositional" generalization from theoretical perspective.

**Reproducibility:**

3: Could reproduce the results with some difficulty. The settings of parameters are underspecified or subjectively determined; the training/evaluation data are not widely available.

**Reviewer Confidence:**

3: Pretty sure, but there's a chance I missed something. Although I have a good feel for this area in general, I did not carefully check the paper's details, e.g., the math, experimental design, or novelty.

---

> ### Author Rebuttal · Authors · 2023-08-28
>
> We deeply thank the reviewer for their invaluable comments, recommendations, and insightful critiques. We are delighted to receive their recognition of our paper's significance in tackling a critical challenge and enhancing model performance in compositional generalization tasks.
>
> We offer our responses to your comments and inquiries below:
>
> > The dataset cartography used in this paper seems to be another way to identify the hard examples, and the way authors utilize it is mainly about curriculum learning (easy-to-hard). I didn't fully get how this is related to compositional generalization. The compositionality is not easily shown in the cartography, and we could not simply assume hard examples are those composite data.
>
> The concept of dataset cartography, as we’ve utilized it, involves arranging training examples on a Cartesian plane based on their hardness and ambiguity levels. This procedure results in the emergence of distinct regions representing easy-to-learn, ambiguous, and hard-to-learn examples. While it's true that the dataset cartography is not directly related to compositionality, our hypothesis is rooted in the idea that certain training examples might hold greater importance than others in achieving better compositional generalization.  Drawing inspiration from Swayamdipta et al. (2020), who revealed that training models on ambiguous subsets enhances out-of-distribution (OOD) generalization, and given that compositional generalization can be seen as an OOD task, we conjectured that dataset cartography might also yield benefits. Although our theoretical understanding is limited, the substantial empirical performance improvements we observed remain noteworthy. While we agree with the reviewer's call for deeper theoretical exploration, we believe our paper provides a comprehensive account of our findings, reserving theoretical analysis for future investigations.
>
> > Why the cartography + curriculum learning could improve compositional generalization?
>
> Initially, our hypothesis leaned toward the idea that ambiguous examples could boost OOD generalization. However, our empirical results shifted this perspective, revealing that hard-to-learn examples contribute more to enhanced generalization accuracy. We speculate that initiating training with hard-to-learn examples prompts the model to frequently encounter these challenging instances during training, leading to better adaptation and performance gains, analogous to training with hard-to-learn subsets.
>
> > What if we simply train the model with enough time and regularization?
>
> Our choice of training setup and repository was inspired by [Csordás et al., EMNLP 2021](https://aclanthology.org/2021.emnlp-main.49), who demonstrated that standard Transformer configurations struggle in compositional generalization. They highlighted that extending training duration and removing early stopping aids in improved generalization. Our experiments adhere to their hyperparameter configurations, ensuring prolonged training with regularization. All models across different datasets underwent the same number of training steps, ensuring fairness in comparison.
>
> > What if we use some simple metric to identify the hard example (e.g., training acc) combined with curriculum learning?
>
> Our approach involves utilizing __the BLEU metric__, which __offers a finer-grained evaluation compared to training accuracy, especially for sequence generation tasks__. Specifically, in our BLEU-based hard-to-learn curriculum learning setup, the model initially encounters the most challenging examples, followed by a gradual unlocking of less complex instances. This rationale extends to both easy-to-learn and ambiguous curriculum learning setups, encapsulating a balanced representation of various example types throughout training.
>
> In summary, we extend our heartfelt gratitude to the reviewer for their invaluable input, which has significantly enriched our paper. While the relationship between dataset cartography and compositional generalization might not be immediately intuitive, our empirical findings demonstrate the efficacy of our approach in enhancing model performance. By combining dataset cartography and curriculum learning, we offer a unique perspective on addressing compositional generalization challenges, underscoring the potential benefits of focusing on hard-to-learn examples. We appreciate the reviewer's thought-provoking questions and have incorporated their insights to strengthen the clarity and impact of our contribution.

---

### Official Review · Reviewer_bjGd · 2023-08-08

**Typos Grammar Style And Presentation Improvements:** 1. Legends in figures 3, 4, 5, and 6 …
**Soundness:** 4

**Excitement:**

3: Ambivalent: It has merits (e.g., it reports state-of-the-art results, the idea is nice), but there are key weaknesses (e.g., it describes incremental work), and it can significantly benefit from another round of revision. However, I won't object to accepting it if my co-reviewers champion it.

**Paper Topic And Main Contributions:**

This work explores methods to improve the compositional generalization of neural sequence generation models. The primary approach leverages dataset cartography (Swayamdipta et al., 2020), extending its original use in out-of-distribution generalization to compositional generalization under sequence generation settings. By redefining confidence and variability for sequence generation, the paper identifies hard-to-learn, ambiguous, and easy-to-learn data points. Experiments demonstrate improvements in compositional generation using two synthetic datasets. Notably, starting the curriculum with hard-to-learn samples led to better final performance, which contradicts common beliefs about curriculum learning.

**Questions For The Authors:**

1. The paper mentions "While curriculum learning also contributes to performance enhancement, its impact appears to be comparatively smaller compared to the use of dataset cartography for subset selection" (L491), which figure is this based on?

**Reasons To Accept:**

1. The paper offers interesting results, challenging common assumptions about curriculum learning and showing that starting with hard-to-learn samples might yield better performance.
2. The paper is well-written and easy to understand.

**Reasons To Reject:**

1. The scope of this work is narrow, focusing only on compositional generalization on synthetic tasks (and specifically for transformer architectures). Extending the analysis to other types of distribution shifts or a broader range of tasks and models would enhance the paper's impact.
2. Methodologically, this paper primarily relies on applying Swayamdipta et al.'s method, with limited originality.
3. Empirically, the experiments are restricted to synthetic datasets, limiting the relevance of the findings to real-world scenarios. The conclusions might be specific to these synthetic cases. For example, even though the conclusions made by this paper --- that hard-to-learn samples are more useful whereas Swayamdipta et al shows hard-to-learn samples are mostly labeling errors, it might be simply due to the fact that in this work synthetic datasets are used which doesn't contain labeling errors.
4. In the curriculum learning experiments, the paper lacks detailed analysis to differentiate between the benefits derived from using hard-to-learn samples and those from curriculum learning itself. Removal of the distributional shift from training to test might clarify the curriculum learning effects.


**Reproducibility:**

4: Could mostly reproduce the results, but there may be some variation because of sample variance or minor variations in their interpretation of the protocol or method.

**Reviewer Confidence:**

4: Quite sure. I tried to check the important points carefully. It's unlikely, though conceivable, that I missed something that should affect my ratings.

---

> ### Author Rebuttal · Authors · 2023-08-28
>
> We express our sincere gratitude to the reviewer for their valuable insights, constructive criticisms, and valuable suggestions. We are pleased to learn that the reviewer recognizes the well-structured nature of our paper and acknowledges the novelty of our approach that challenges established assumptions about curriculum learning.
>
> In response to the provided comments and questions:
>
> > The scope of this work is narrow, focusing only on compositional generalization on synthetic tasks (and specifically for transformer architectures). Extending the analysis to other types of distribution shifts or a broader range of tasks and models would enhance the paper's impact.
>
> Acknowledging this valid concern, we have extended our evaluation to include experiments conducted on the SMCalFlow-CS dataset  [(Yin et al., NAACL 2021)](https://aclanthology.org/2021.naacl-main.225), a natural dataset utilized for analyzing systematic generalization [(Levy et al., ACL 2023)](https://aclanthology.org/2023.acl-long.78). It is worth noting that due to time constraints, we were able to conduct experiments on only 2 out of the dataset's 5 splits. Nevertheless, our findings from these experiments align harmoniously with the conclusions outlined in our paper, thereby reinforcing the robustness of our findings on real-world datasets as well. The outcomes of these supplementary experiments are detailed below:
>
> __SM-Calflow-CS - 16 Split (50% train set)__
> | | InvPPL | CHIA | BLEU |
> |-------|----------:|-------------:|------:|
> |easy-to-learn | 0.0% | 0.0% | 0.0% |
> |ambiguous | 0.0% | 2.0% | 2.3% |
> |hard-to-learn | **4.5%** | **4.5**% | **6.8%** |
> |random | 0.4% | 0.4% | 0.4% |
> |100% train set| 4.2% | 4.2% | 4.2% |
>
> ***
>
> __SM-Calflow-CS - 32 Split (50% train set)__
>
> | | InvPPL | CHIA | BLEU |
> |-------|----------:|-------------:|------:|
> |easy-to-learn | 0.1% | 0.0% | 1.5% |
> |ambiguous | 7.0% | 7.8% | 11.6% |
> |hard-to-learn | **16.8%** | **17.5%** | **15.9%** |
> |random | 5.9% | 5.9% | 5.9% |
> |100% train set| 15.6% | 15.6% | 15.6% |
>
> Furthermore, we expanded our investigation by applying dataset cartography to a Bi-LSTM with attention model on the COGS dataset. The outcomes of these new experiments are given below:
>
> __COGS - Bi-LSTM (50% train set)__
> | | InvPPL | CHIA | BLEU |
> |-------|----------:|-------------:|------:|
> |easy-to-learn | 2.7% | 8.3% | 9.9% |
> |ambiguous | 14.2% | 2.8% | 11.9% |
> |hard-to-learn | **16.6%** | **15.7%** | **20.2%** |
> |random | 9.9% | 9.9% | 9.9% |
> |100% train set | 13.7% | 13.7% | 13.7% |
>
> Notably, these results provide additional confirmation that hard-to-learn subsets consistently yield superior results compared to training on the entire dataset, thereby reinforcing the claims articulated in our main paper.
>
> In summary, these supplementary results indicate the transferability of our claims to real-world datasets and alternative model architectures. However, we acknowledge the need for further experiments to achieve a more comprehensive and conclusive view.
>
> > Methodologically, this paper primarily relies on applying Swayamdipta et al.'s method, with limited originality.
>
> We respectfully disagree with this assessment. Our originality lies in the introduction and thorough analysis of novel seq2seq cartography metrics, namely inverse PPL and BLEU metrics. Moreover, we novelly investigate compositional generalization through training dynamics. This unique approach adds depth and insights to our understanding of systematic generalization.
>
> > Empirically, the experiments are restricted to synthetic datasets, limiting the relevance of the findings to real-world scenarios. The conclusions might be specific to these synthetic cases. For example, even though the conclusions made by this paper --- that hard-to-learn samples are more useful whereas Swayamdipta et al shows hard-to-learn samples are mostly labeling errors, it might be simply due to the fact that in this work synthetic datasets are used which doesn't contain labeling errors.
>
> While our initial experiments focused on synthetic datasets, we also conducted additional experiments using the real-world SMCalFlow-CS dataset. These supplementary experiments lend support to the applicability of dataset cartography within the context of natural compositional generalization datasets. This indicates that the insights drawn from our work can be extended beyond synthetic datasets.
>
> While we concur that hard-to-learn subsets may include labeling errors, as evidenced in (Swayamdipta et al., 2020), our empirical findings show that other properties of hard-to-learn subsets contribute to performance improvement, even in natural compositional generalization datasets.
>
> > In the curriculum learning experiments, the paper lacks detailed analysis to differentiate between the benefits derived from using hard-to-learn samples and those from curriculum learning itself.
>
> We would like to clarify that curriculum learning mainly outlines the unlocking protocol for the full training dataset. The unlocking order, however, is solely determined by dataset cartography. While Hacohen and Weinshal (2019) employ exponential unlocking, Zhang et al. (2019) follow a linear unlocking strategy, with additional trivialities.
>
> > Removal of the distributional shift from training to test might clarify the curriculum learning effects.
>
> Due to the inherent nature of compositional generalization, a strict distributional shift exists between training subsets and test sets. Test sets consistently involve novel compositions not encountered during training, resulting in an inevitable distributional shift. Thus, we respectfully differ with the reviewer’s viewpoint on this matter.
>
> > The paper mentions "While curriculum learning also contributes to performance enhancement, its impact appears to be comparatively smaller compared to the use of dataset cartography for subset selection" (L491), which figure is this based on?
>
> While we don’t have a dedicated figure for a direct comparison, through a juxtaposition of Figures 3-4 and Figures 5-6 with Tables 2-3, it becomes apparent that curriculum learning yields a performance improvement of no more than a 4.5% for CFQ and up to 10% for COGS.
>
> Regarding the typos and presentation improvements:
> > Legends in figures 3, 4, 5, and 6 are too small, hindering comprehension of the curriculum learning section.
>
> > L094: Replace "originally used in (Swayamdipta et al., 2020)" with "originally used in Swayamdipta et al. (2020)."
>
> > L111: Modify to "quantifying how hard a sequence is, is not straightforward."
>
> We addressed these typographical concerns and presentation issues in the revised version.
>
> In summary, we extend our deep gratitude to the reviewer for their insightful feedback. We've broadened our analysis to incorporate real-world datasets and alternative model architectures, reinforcing the transferability of our findings. While drawing inspiration from prior work, our approach introduces novel perspectives and metrics, enriching our understanding of compositional generalization. The valuable critiques raised by the reviewer have been instrumental in refining our paper, and we are committed to addressing their suggestions to enhance the quality of our contribution.

---

### Official Review · Reviewer_1Pgd · 2023-08-11

**Soundness:** 4

**Excitement:**

3: Ambivalent: It has merits (e.g., it reports state-of-the-art results, the idea is nice), but there are key weaknesses (e.g., it describes incremental work), and it can significantly benefit from another round of revision. However, I won't object to accepting it if my co-reviewers champion it.

**Paper Topic And Main Contributions:**

This paper explores how to use dataset cartography to improve compositional generalization. It studies different measures (Inv PPL, CHIA, BLEU) for assessing confidence and variability in semantic parsing tasks. Furthermore, it provides a detailed analysis of the impacts of different types of examples. Finally, the paper uses dataset cartography as a curriculum learning and sampling criterion, and demonstrates its effectiveness in improving compositional generalization.

### Updates after rebuttal
- Increased "soundness" from "good" to "strong"

**Questions For The Authors:**

A. There are a few literatures that study the impacts of example selections on compositional generalization in the in-context learning setting, e.g. Levy et al, Gupta et al. Although these are technically different from what was studied in this paper, I wonder if the authors have an intuition or have investigated whether sampling hard-to-learn examples could also be beneficial for an in-context learning setup.

B. Have the authors considered non-synthetic compositional generalization datasets? It would be nice to see a study on those, even just for small datasets such as GeoQuery.

C. Do the authors have an explanation about the inconsistency of CL performance even within the dataset? For example, “no curriculum” significantly outperforms others in Figure 3 but not for Figure 4 and COGS dataset.

Gupta et al. Structurally Diverse Sampling for Sample-Efficient Training and Comprehensive Evaluation

**Reasons To Accept:**

- Provide an interesting aspect of leveraging training dynamics to improve compositional generalization, which has not been well explored before.
- The experiments and analyses are comprehensive and well-designed.
- Well written, easy to follow. The visualizations are clear.

**Reasons To Reject:**

- The paper only studies one split from each of two synthetic datasets. It’s hard to know whether the conclusions can be translated to other splits and datasets.
- The effectiveness of leveraging dataset cartography for CL is unclear. In most cases, no curriculum appears to perform better or is on par with a strategy that starts the curriculum with hard-to-learn samples.

**Reproducibility:**

4: Could mostly reproduce the results, but there may be some variation because of sample variance or minor variations in their interpretation of the protocol or method.

**Reviewer Confidence:**

4: Quite sure. I tried to check the important points carefully. It's unlikely, though conceivable, that I missed something that should affect my ratings.

**Typos Grammar Style And Presentation Improvements:**

- L95, L104, L292, L322, etc. \citet{Swayamdipta et al., 2020}

---

> ### Author Rebuttal · Authors · 2023-08-28
>
> We sincerely thank the reviewer for their invaluable feedback and insightful suggestions. We greatly appreciate the acknowledgment of our approach in tackling the compositional generalization problem through the lens of training dynamics. Additionally, we are thankful for the positive remarks concerning our experimental setup's comprehensiveness, the clarity of our visualization, and the overall quality of the writing.
>
> Addressing the comments and questions provided:
>
> > The paper only studies one split from each of two synthetic datasets. It’s hard to know whether the conclusions can be translated to other splits and datasets.
>
> > Have the authors considered non-synthetic compositional generalization datasets? It would be nice to see a study on those, even just for small datasets such as GeoQuery.
>
> We acknowledge this pivotal concern, which we have thoughtfully addressed within the Limitations section of our paper. While we concentrated our efforts on two widely employed compositional generalization datasets, we recognize the importance of ensuring the transferability of our findings to other datasets. To address this, __we conducted experiments on a natural compositional semantic parsing dataset, SM-Calflow-CS [(Yin et al., NAACL 2021)](https://aclanthology.org/2021.naacl-main.225)__, which has been utilized in prior research [(Levy et al., ACL 2023)](https://aclanthology.org/2023.acl-long.78).
>
> Despite time constraints limiting us to experiments on only 2 out of the dataset's 5 splits, we discovered that the insights drawn from these experiments were consistent with the observations presented in our paper. This consistency lends additional support to the robustness of our findings.
>
> The outcomes of these experiments have been included in the following tables, demonstrating that, consistent with our main paper, hard-to-learn subsets outperform 100% dataset training:
>
> __SM-Calflow-CS - 16 Split (50% train set)__
> | | InvPPL | CHIA | BLEU |
> |-------|----------|:-------------:|------:|
> |easy-to-learn | 0.0% | 0.0% | 0.0% |
> |ambiguous | 0.0% | 2.0% | 2.3% |
> |hard-to-learn | **4.5%** | **4.5**% | **6.8%** |
> |random | 0.4% | 0.4% | 0.4% |
> |100% train set| 4.2% | 4.2% | 4.2% |
>
> ***
>
> __SM-Calflow-CS - 32 Split (50% train set)__
>
> | | InvPPL | CHIA | BLEU |
> |-------|----------|:-------------:|------:|
> |easy-to-learn | 0.1% | 0.0%% | 1.5% |
> |ambiguous | 7.0% | 7.8% | 11.6% |
> |hard-to-learn | **16.8%** | **17.5%** | **15.9%** |
> |random | 5.9% | 5.9% | 5.9% |
> |100% train set| 15.6% | 15.6% | 15.6% |
>
> We firmly believe that these supplementary experiments bolster the generality of our conclusions, providing a more comprehensive understanding of the dynamics underlying compositional generalization.
>
> > The effectiveness of leveraging dataset cartography for CL is unclear. In most cases, no curriculum appears to perform better or is on par with a strategy that starts the curriculum with hard-to-learn samples.
>
> > Do the authors have an explanation about the inconsistency of CL performance even within the dataset? For example, “no curriculum” significantly outperforms others in Figure 3 but not for Figure 4 and COGS dataset.
>
> We agree with the reviewer's observation, particularly in the context of the CFQ dataset. While certain improvements are evident in the COGS dataset (Figures 3 and 4), we acknowledge the valid concerns raised about the effectiveness of curriculum learning (CL) on the CFQ dataset. The inconsistency in CL performance indeed stems from the variation in designs of the curricula, as shown between Figures 3-4 and Figures 5-6.
>
> Our methodology employs dataset cartography as the criterion for establishing the unlocking sequence (ranging from easy to hard-to-learn, vice versa, and from most- to least-ambiguous), while the CL framework governs the unlocking pace. This discrepancy is apparent in the exponential (approx. 2% → 4% → … → 100%) and linear (10% → 20% → … → 100%) mechanisms illustrated in Figures 3 and 4, respectively. Moreover, nuances such as arranging mini-batches with similar levels of hardness contribute to the divergent outcomes. Hence, the variability in CL performance emerges from the intricate interplay of these factors, and we intend to clarify this in our revised manuscript.
>
> Importantly, it's worth noting that our primary aim with CL experiments was to offer a more comprehensive analysis of training dynamics in compositional generalization, rather than to introduce a practical CL strategy. The evaluated CL strategies serve to demonstrate the significance of incorporating hard-to-learn, easy-to-learn, and ambiguous samples during training within the context of systematic generalization. We appreciate the reviewer for highlighting this aspect, and we will make the necessary clarifications to convey this point effectively in our paper.
>
> > There are a few literatures that study the impacts of example selections on compositional generalization in the in-context learning setting, e.g. Levy et al, Gupta et al. Although these are technically different from what was studied in this paper, I wonder if the authors have an intuition or have investigated whether sampling hard-to-learn examples could also be beneficial for an in-context learning setup.
>
> Following the submission deadline, __we have indeed contemplated potential applications of dataset cartography in an in-context learning setup__. We envision that identifying hard-to-learn examples using a small language model and utilizing them to select in-context examples for a larger language model could be a promising future direction. This idea aligns well with the reviewer's suggestion, and we value their insightful input. We have plans to further investigate this matter in the near future.
>
> Addressing the typo and presentation improvements:
> > L95, L104, L292, L322, etc. \citet{Swayamdipta et al., 2020}
>
> We appreciate the reviewer for pointing out these typos, and we corrected them in our revised version.
>
> In conclusion, the reviewer's insights have been instrumental in revising our paper, and we are deeply appreciative of their thorough assessment. We are dedicated to addressing all the raised concerns and making the necessary improvements to ensure the clarity, rigor, and impact of our work.

---

### Meta-Review · Area_Chair_orJ4 · 2023-09-19

**Recommendation:** 3

**Metareview:**

This paper explores how to use dataset cartography to improve compositional generalization. In short, the paper uses dataset cartography as a curriculum learning and sampling criterion, and demonstrates its effectiveness in improving compositional generalization.

Overall, I think the paper proposes an exciting direction for the community to look at. On the other hand, there are some drawbacks of the paper: (1)  The scope of this work is narrow, (2) the found hard examples are not necessarily compositional examples. More investigation needs to be done to understand it.

Based on the merits and limitations of the paper, I would recommed acceptance to findings.

---

### Decision · Program_Chairs · 2023-10-07

**Decision:**

Accept-Findings

**Comment:**

This paper explores how to use dataset cartography to improve compositional generalization. In short, the paper uses dataset cartography as a curriculum learning and sampling criterion, and demonstrates its effectiveness in improving compositional generalization.

Overall, I think the paper proposes an exciting direction for the community to look at. On the other hand, there are some drawbacks of the paper: (1)  The scope of this work is narrow, (2) the found hard examples are not necessarily compositional examples. More investigation needs to be done to understand it.

Based on the merits and limitations of the paper, I would recommed acceptance to findings.